## TOPICAL REVIEW

# Mitochondrial carbonic anhydrase VA and VB: properties and roles in health and disease

Ashok Aspatwar[1,2] 🆔, Claudiu T. Supuran[3], Abdul Waheed[4], William S. Sly[4] and Seppo Parkkila[1,2] 🆔

[1]*Faculty of Medicine and Health Technology, Tampere University, Tampere, Finland*
[2]*Fimlab Ltd and Tampere University Hospital, Tampere, Finland*
[3]*Neurofarba Department, Sezione di Chimica Farmaceutica e Nutraceutica, Università degli Studi di Firenze, Sesto Fiorentino, Firenze, Italy*
[4]*Department of Biochemistry and Molecular Biology, Edward A. Doisy Research Center, Saint Louis University School of Medicine, St Louis, MO, USA*

Handling Editors: Laura Bennet & Kyle McCommis

The peer review history is available in the Supporting information section of this article (https://doi.org/10.1113/JP283579#support-information-section).

**Abstract**   Carbonic anhydrase V (CA V), a mitochondrial enzyme, was first isolated from guinea-pig liver and subsequently identified in mice and humans. Later, studies revealed that the mouse genome contains two mitochondrial CA sequences, named *Car5A* and *Car5B*. The CA VA enzyme is most highly expressed in the liver, whereas CA VB shows a broad tissue distribution. *Car5A* knockout mice demonstrated a predominant role for CA VA in ammonia detoxification, whereas the roles of CA VB in ureagenesis and gluconeogenesis were evident only in the absence of CA VA. Previous studies have suggested that CA VA is mainly involved

**Ashok Aspatwar** is adjunct professor of molecular biology at the Faculty of Medicine and Health Technology, Tampere University, Finland. He obtained a master's degree in bioinformatics in 2011 and completed a PhD degree in 2014. During his doctoral research, he studied the roles of carbonic anhydrase-related proteins (CARPs) in zebrafish using functional genomics approaches. His current research focus is on understanding the roles of carbonic anhydrases in persistence of *Mycobacterium tuberculosis* in the host. **Seppo Parkkila** is dean and professor of anatomy at the Faculty of Medicine and Health Technology, Tampere University, Finland. He obtained an MD degree in 1991 and PhD in 1994. In 1996−1998, he was a visiting researcher at Saint Louis University. In 2002, he became full professor at Tampere University. In 2016−2018, he served as vice-rector of the university. His research is focused on functional genomics of carbonic anhydrases and pH regulation.

 

in the provision of $HCO_3^-$ for biosynthetic processes. In children, mutations in the *CA5A* gene led to reduced CA activity, and the enzyme was sensitive to increased temperature. The metabolic profiles of these children showed a reduced supply of $HCO_3^-$ to the enzymes that take part in intermediary metabolism: carbamoylphosphate synthetase, pyruvate carboxylase, propionyl-CoA carboxylase and 3-methylcrotonyl-CoA carboxylase. Although the role of CA VB is still poorly understood, a recent study reported that it plays an essential role in human Sertoli cells, which sustain spermatogenesis. Metabolic disease associated with CA VA appears to be more common than other inborn errors of metabolism and responds well to treatment with *N*-carbamyl-ʟ-glutamate. Therefore, early identification of hyperammonaemia will allow specific treatment with *N*-carbamyl-ʟ-glutamate and prevent neurological sequelae. Carbonic anhydrase VA deficiency should therefore be considered a treatable condition in the differential diagnosis of hyperammonaemia in neonates and young children.

(Received 17 August 2022; accepted after revision 30 November 2022; first published online 4 December 2022)

**Corresponding author** Seppo Parkkila: Faculty of Medicine and Health Technology, Tampere University, Tempere FI-33520, Finland. Email: seppo.parkkila@tuni.fi

**Abstract figure legend** Carbonic anhydrase isozymes VA and VB are mitochondrial enzymes that contribute to several physiological functions, mainly in intermediary metabolism. The liver hepatocytes are the main source of carbonic anhydrase VA, with weaker signals in the brain, testis and muscle. The VB isozyme is more widely spread in several organs, such as brain, heart, liver, lung, kidney, spleen, intestine, testis, muscle and pancreas.

## Introduction

The carbonic anhydrases (CAs), EC 4.2.1.1, are metalloenzymes that occur abundantly in nature and are found in almost all organisms that have been studied (Aspatwar et al., 2010). The fundamental role of CAs is to catalyse the simple physiological reaction of reversible hydration of $CO_2$ to generate $HCO_3^-$ and $H^+$ ions (Aspatwar et al., 2022). The activity of mammalian CAs modulates cellular processes associated with respiration and transport of $CO_2$, providing $HCO_3^-$ as a substrate for biosynthetic pathways. Thus, CAs take part in many important physiological reactions, such as pH regulation, electrolyte and fluid secretion, biosynthetic reactions, bone resorption, calcification, tumorigenicity and many other physiological or pathological processes (Aspatwar et al., 2022; Supuran, 2008).

In mammals, 13 different $\alpha$-CA isozymes and three CA-related proteins have been identified and characterized (Aspatwar et al., 2014, 2010; Supuran, 2008). The 13 mammalian CA isozymes differ in subcellular localization, expression levels, kinetic properties and sensitivity to different inhibitors (Supuran, 2008). Of the 13 catalytically active CAs, five are cytosolic (CA I–III, CA VII and CA XIII), five are membrane-bound isozymes (CA IV, CA IX, CA XII, CA XIV and CA XV), two are mitochondrial forms (CA VA and CA VB) and one is a secreted isozyme (CA VI). Among the membrane-bound CAs, CA IV and XV are anchored to membranes by means of glycosylphosphatidylinositol tails, whereas isozymes IX, XII and XIV are transmembrane proteins with one transmembrane domain. Interestingly, all membrane-bound isozymes have their active site outside the cell and are therefore known as extracellular CAs (Supuran, 2008; Supuran & Scozzafava, 2000). The three CA-related proteins, known as CARP VIII, X and XI, are catalytically inactive owing to the lack of one or more of the three histidine residues that are required for the coordination of the Zn(II) atom within the active site (Aspatwar et al., 2010, 2013, 2014).

Interestingly, CA VA and CA VB are the only mammalian isoforms that are localized in mitochondria (Dodgson et al., 1980; Shah et al., 2000). The presence of mitochondrial CA was first reported in the rat liver and kidney in 1959, and mitochondrial CA V was isolated from guinea-pig liver and characterized in 1980 (Datta & Shepard, 1959; Dodgson et al., 1980). The mitochondrial CA was named CA V in the order of discovery, because the existence of another mitochondrial CA, now called CA VB, was not known at the time. Henceforth, in the text, the previously discovered mitochondrial CA V is referred to as CA VA. Mitochondrial CA VA has been shown to be involved in several important biosynthetic processes, such as ureagenesis, gluconeogenesis and lipogenesis (Dodgson, 1987; Dodgson & Cherian, 1989; Hazen et al., 1996; Lynch et al., 1995). Indeed, $HCO_3^-$ is the key substrate for pyruvate carboxylase (PC), acetyl-CoA carboxylase (ACC) and carbamoyl phosphate synthetases (CPS) I and II (Allred & Reilly, 1996; Attwood, 1995; Hazen et al., 1996). In biosynthetic processes, CA VA provides $HCO_3^-$ ions for the first step in the urea cycle, catalysed by CPS I, and for the first step of

gluconeogenesis, whereby PC converts pyruvate into oxaloacetate (Aspatwar et al., 2022; Dodgson, 1991; Dodgson et al., 1983; Hazen et al., 1996; Metcalfe et al., 1985).

The second mitochondrial enzyme, CA VB, was isolated from pancreas and salivary glands and was found to be homologous to CA VA (Fujikawa-Adachi et al., 1999; Shah et al., 2000). Both isoforms have a distinct tissue distribution (Saarnio et al., 1999; Shah et al., 2000). Carbonic anhydrase VA is expressed mainly in the liver, with some expression also observed by Western blot in the brain, testis and skeletal muscle (Shah et al., 2000). The CA VB isozyme has a much wider tissue distribution and is expressed in the heart, liver, lung, spleen, intestine, pancreas, testis, skeletal muscle, kidney, salivary gland, brain and spinal cord, suggesting different physiological roles for these two mitochondrial isozymes (Fujikawa-Adachi et al., 1999; Ghandour et al., 2000; Shah et al., 2000).

Studies on the roles of mitochondrial CA VA and CA VB using an electrochemical method of wired mitochondria have shown that these CAs have important functions in the regulation of metabolism (Arechederra et al., 2013). Inhibition of CA VA and CA VB by isoform-specific sulfonamides has a dramatic effect on pyruvate metabolism, followed by fatty acid and succinate metabolism (Arechederra et al., 2013). Recent studies have shown that inhibition of CA activity in human Sertoli cells (hSCs), rich in CA VB, affects the expression of genes involved in mitochondrial biogenesis and lipid metabolism in these cells (Bernardino et al., 2019). Carbonic anhydrase activity might be required for normal metabolism in hSCs and might play an essential role in spermatogenesis (Bernardino et al., 2019). In addition, *Car5A* and *Car5B* knockout mouse studies showed that the absence of *Car5A* leads to hyperammonaemia and poor growth, suggesting that the CA VA enzyme is required for detoxification of ammonia in the liver (Shah et al., 2013). Interestingly, *Car5B* null mice showed normal growth and normal ammonia levels, whereas *Car5A* and *Car5B* double knockout mice showed more severely impaired growth with greater hyperammonaemia, suggesting a mutual contribution of both CA VA and CA VB in detoxification of ammonia (Shah et al., 2013). In humans, the absence of enzymatically active CA VA owing to *CA5A* gene mutations leads to hyperammonaemia in early childhood, confirming the role of CA VA in ammonia detoxification, which is resolved with the administration of carglumic acid (Diez-Fernandez et al., 2016; van Karnebeek et al., 2014).

In this review, we present comprehensive information on the molecular biology, biochemistry and physiological functions of CA VA and CA VB and their roles in health and disease. In addition, we present data related to enzyme

**Table 1. Catalytic activities of human carbonic anhydrases for $CO_2$ hydration reaction**

| Isozyme | $k_{cat}$ $(s^{-1})$ | $K_m$ (mM) | $k_{cat}/K_m$ $(M^{-1}\ s^{-1})$ |
|---|---|---|---|
| CA I | $2.0 \times 10^5$ | 4.0 | $5.0 \times 10^7$ |
| CA II | $1.4 \times 10^6$ | 9.3 | $1.5 \times 10^8$ |
| CA III | $1.3 \times 10^4$ | 52.0 | $2.5 \times 10^5$ |
| CA IV | $1.1 \times 10^6$ | 21.5 | $5.1 \times 10^7$ |
| CA VA | $2.9 \times 10^5$ | 10.0 | $2.9 \times 10^7$ |
| CA VB | $9.5 \times 10^5$ | 9.7 | $9.8 \times 10^7$ |
| CA VI | $3.4 \times 10^5$ | 6.9 | $4.9 \times 10^7$ |
| CA VII | $9.5 \times 10^5$ | 11.4 | $8.3 \times 10^7$ |
| CA IX* | $1.1 \times 10^6$ | 7.5 | $1.5 \times 10^8$ |
| CA XII | $4.2 \times 10^5$ | 12.0 | $3.5 \times 10^7$ |
| CA XIII | $1.5 \times 10^5$ | 13.8 | $1.1 \times 10^7$ |
| CA XIV | $3.1 \times 10^5$ | 7.9 | $3.9 \times 10^7$ |

*Carbonic anhydrase (CA) domain + proteoglycan domain.

inhibition studies using different classes of CA inhibitors from *in vitro* experiments.

**Biochemistry of CA VA and CA VB.** The activity of CA VA, located in the mitochondrial matrix of guinea-pig liver, was measured by monitoring the kinetics of the disappearance of $^{18}O$ from $C^{16}O^{18}O$ at chemical equilibrium in bicarbonate buffer as $^{18}O$ exchanges with $^{16}O$ in water (Dodgson et al., 1982). These measurements of the liver mitochondrial enzyme demonstrated distinct CA activity and showed that it is sensitive to inhibition by nanomolar concentrations of acetazolamide (AAZ) (Dodgson et al., 1983, 1982; Hilvo et al., 2008; Itada & Forster, 1977). The activity of CA VB was measured using extracts of COS-7 cells along with human CA II, and the extract containing CA VB showed low but significant catalytic activity compared with human CA II. Inhibition studies using AAZ showed a 75% reduction in the catalytic activity of CA VB, similar to human CA II (Aspatwar et al., 2022). Detailed catalytic activities of all human CAs for the $CO_2$ hydration reaction have been published, and the results are shown in Table 1 (Hilvo et al., 2008). Carbonic anhydrase VA showed moderate catalytic activity ($k_{cat}$ = $2.9 \times 10^5$ $s^{-1}$, $K_m$ = 10 mM and $k_{cat}/K_m$ = $2.9 \times 10^7$ $M^{-1}\ s^{-1}$) that was similar to the activity levels of human CA I, VI, XII, XIV and murine XV isozymes (Hilvo et al., 2008).

Later, studies were carried out on the effects of pH on the structure, function and stability of purified recombinant human CA VA using spectroscopic techniques, such as circular dichroism, fluorescence and absorbance measurements, over a range of pH from 2.0 to 11.5 (Idrees et al., 2016). The CA VA enzyme aggregates at acidic pH (2.0–5.0). It maintains its secondary structure

**Table 2. Details of the human carbonic anhydrase VA and VB sequences**

| CA isoform | Complementary DNA*length | Amino acids[†] | MW[‡] | Location | Ensembl ID | NCBI RefSeq | References |
|---|---|---|---|---|---|---|---|
| hCA VA | 1123 | 305 | 30 | Chromosome 16: 16q24.3 | ENST00000649794.3 | NM_0 01739.2 | Nagao et al. (1995, 1993) |
| hCA VB | 1182 | 317 | 32 | Chromosome X: Xp22.1 | ENST00000318636.8 | NM_0 07220.4 | Fujikawa-Adachi et al. (1999) |

Abbreviations: CA, carbonic anhydrase; hCA, human carbonic anhydrase.
[*]Length of the complementary DNA (in base pairs).
[†]Full-length sequences.
[‡]Molecular weight of processed sequences.

at pH 7.0–11.5 and shows appreciable catalytic activity at neutral to alkaline pH ($7.0 < \text{pH} \leq 11.5$), exhibiting maximum activity at pH 9.0. Mitochondria are dynamic organelles, with tightly regulated pH in different intra-mitochondrial compartments. Although it is well known that the function of an enzyme is dependent on the pH of the environment (Talley & Alexov, 2010), CA VA and CA VB are the only isoforms of the CA family that are present in a highly alkaline environment (pH 8.5) (Stoll & Blanchard, 1990; Supuran, 2008).

**Molecular biology of CA VA and CA VB.** Carbonic anhydrase VA was first purified and its N-terminal sequence information determined from guinea-pig and rat liver mitochondria (Dodgson, 1991; Nagao et al., 1994); later, mouse complementary DNA (cDNA) was isolated from a liver cDNA library (Amor-Gueret & Levi-Strauss, 1990). Subsequently, a full-length cDNA clone encoding human CA VA from a human liver cDNA library was isolated by Nagao et al. (1993). The human *CA5A* cDNA contained 1123 bp, including a 55 bp 5′ untranslated region at the 5′ region, a 915 bp open reading frame and a 153 bp untranslated region at the 3′ end (Nagao et al., 1993). The 48.52 kb human *CA5A* gene is located on chromosome 16q24.3 and contains seven exons and six introns, and the positions of the exon–intron boundaries are identical to those of the human *CA2*, *CA3* and *CA7* genes (Table 2 and Fig. 1; Nagao et al., 1995).

Expression of CA VA in COS-7 cells produced an active enzyme with 34 kDa precursor and 30 kDa mature enzyme forms. Normal human liver showed only the mature 30 kDa polypeptide band. Analysis of the CA VA purified from COS-7 cells showed that processing of the enzyme involved removal of the 38 amino acid mitochondrial leader sequence at the N-terminal end of the precursor form, resulting in the mature 30 kDa polypeptide. The deduced sequence (267 amino acids) of mature human CA VA showed 30−49% similarity to other human CAs (CA I–VII) and 76% similarity to mouse CA VA.

A clone for human *CA5B* cDNA was isolated from human pancreas and salivary glands containing a putative sequence of 1182 nucleotides that encodes a 317 amino acid protein with a predicted mass of 36.4 kDa (Fujikawa-Adachi et al., 1999). The cDNA sequence contained the presumed first methionine in a consensus Kozak sequence (AAZATGG) (Kozak, 1996). The human *CA5B* gene is located on a ∼50 kb strand of the chromosome Chr Xp22.1 (Table 2 and Fig. 2).

The open reading frame of *CA5B* contains 951 bp, which codes for 317 amino acids. The CA VB primary sequence contains a hydrophobic N-terminal mitochondrial signal sequence of 33 amino acid residues. The relative molecular mass of CA VB is 32 kDa (Table 2). The CA VB amino acid sequence showed the highest degrees of similarity (66%) and identity (64.16%) with CA VA (Fig. 3) (Gavel & von Heijne, 1990a). Carbonic anhydrase VB contains a putative site for

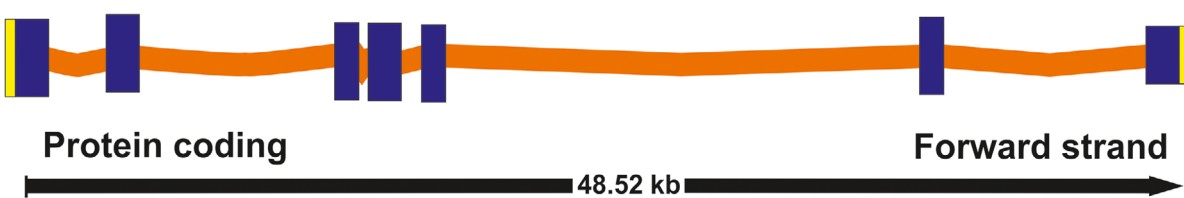

**Figure 1. Structure of the human *CA5A* gene**
The transcript is derived from seven exons (exons shown in violet are coding, and the regions marked with yellow are untranslated parts of exons 1 and 7). The transcript model of the gene was obtained from the Ensembl genome browser (ENST00000649794.3).

N-glycosylation (Asn-Xaa-Thr) at position +257 (residue position is annotated with a '+' before the residue number when protein numbering is not inclusive of the signal peptide), unlike CA VA. Although human CA VA and CA VB contain four and six cysteine residues, respectively, no disulfide bridges as post-translational modifications have been reported in these enzymes (Di Fiore et al., 2020). Subcellular localization analyses using confocal fluorescence microscopy in transfected COS-7 cells showed granular, intracellular signals, consistent with the mitochondrial expression of CA VB (Parkkila et al., 1998).

**Expression of CA VA and CA VB.** Carbonic anhydrase VA was initially detected by biochemical methods in the mitochondria of the rat liver and kidney and in guinea-pig liver and skeletal muscle (Dodgson et al., 1983, 1980). The first detailed Western blotting and immuno-histochemical analyses of mitochondrial CA V expression were performed in human and rat gastrointestinal tract tissues (Saarnio et al., 1999). At that time, mitochondrial CA V was still thought to be a single enzyme. Notably, the two forms, CA VA and VB, were identified as distinct enzymes only a few months later (Fujikawa-Adachi et al., 1999). Immunohistochemical staining showed mitochondrial CA V expression in various segments of the alimentary canal mucosa from the stomach to the rectum (Saarnio et al., 1999). Expression was observed in the parietal cells and gastrin-producing G-cells of the stomach and intestinal enterocytes. These results suggested an important role for the mitochondrial CA in alimentary canal physiology. In another study, the

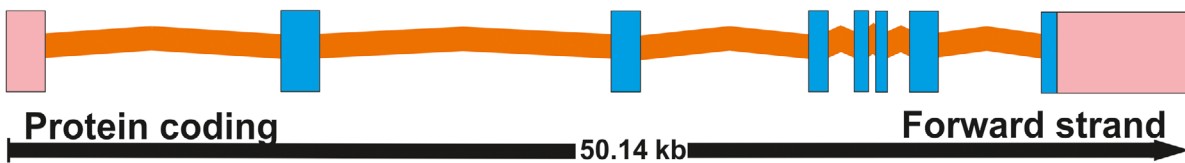

**Figure 2. Structure of the human *CA5B* gene**
It contains eight exons, of which the first exon is untranslated and the last contains a long untranslated part (shown in pink). The transcript length is 6837 bp, and the translated polypeptide consists of 317 residues. The transcript of the gene was obtained from the Ensembl genome browser (ENST00000649794.3).

```
CA VA : MLGRNTWKTSAFSFLVEQMWAPLWSRSMRPGRWCSQRSCAWQTSNNTLHPLWTVPVSVPG  60
CA VB : MVVMNSLRVILQASPGKLLWRKFQIPRFMPARPCSLYTCTYKTRNRALHPLWESVDLVPG  60
        *     *              *       * * **  *     * *  ***** ***

CA VA : GTRQSPINIQWRDSVYDPQLKPLRVSYEAASCLYIWNTGYLFQVEFDDATEASGISGGPL 120
CA VB : GDRQSPINIRWRDSVYDPGLKPLTISYDPASCLHVWNNGYSFLVEFEDSTDKSVIKGGPL 120
        * ******* ******* ****   **  ****  ***** * *** * *  * * ****

CA VA : ENHYRLKQFHFHWGAVNEGGSEHTVDGHAYPAELHLVHWNSVKYQNYKEAVVGENGLAVI 180
CA VB : EHNYRLKQFHFHWGAIDAWGSEHTVDSKCFPAELHLVHWNAVRFENFEDAALEENGLAVI 180
        * ************   ******    **********  *    *    *    *******

CA VA : GVFLKLGAHHQTLQRLVDILPEIKHKDARAAMRPFDPSTLLPTCWDYWTYAGSLTTPPLT 240
CA VB : GVFLKLGKHHKELQKLVDTLPSIKHKDALVEFGSFDPSCLLMTCPDYWTYSGSLTTPPLS 240
        ******* ** *** * ****   **** ** ** ***** *******

CA VA : ESVTWIIQKEPVEVAPSQLSAFRTLLFSALGEEEKMMVNNYRPLQPLMNRKVWASFQATN 300
CA VB : ESVTWIIKKQPVEVDHDQLEQFRTLLFTSEGEKEKRMVDNFRPLQPLMNRTVRSSFRHDY 300
        ******* * ****    **** *****  ** ** ** * ********* *   **

CA VA : EGTRS------------ 305
CA VB : VLNVQAKPKPATSQATP 317
```

**Figure 3. Alignment of the human carbonic anhydrase VA and VB sequences**
The N-termini of both carbonic anhydrase (CA) VA and VB contain hydrophobic domains. The blue arrowheads indicate presumed cleavage sites for the mitochondrial targeting peptides of CA VA (38 amino acid residues) and CA VB (33 residues) (Gavel & von Heijne, 1990b). The mature CA VB protein contains six cysteine residues (red boxes). Two of the six cysteine residues are not observed in CA VA.

immunoblotting of mitochondrial preparations from adipose tissue and liver revealed that both tissues contain similar amounts of CA V (Hazen et al., 1996).

Expression studies of CA VB using Northern blot analyses of poly (A)+ RNA from a panel of human tissues showed transcripts of four different sizes (~1.3, 2.6, 4.4 and 6 kb). The 1.3 kb transcript that corresponds to the full-length *CA5B* cDNA was observed in heart and skeletal muscle. RT–PCR showed positive signals in the pancreas, kidney, salivary glands and spinal cord but not in the liver (Fujikawa-Adachi et al., 1999). In another study, RT–qPCR analyses of CAs showed the presence of mRNAs for three CAs (CA IX, XII and VB) in hSCs. Among them, the expression of *CA5B* mRNA was abundant, and interestingly, no expression of CA VA was found in these cells, suggesting a specific role for CA VB in the regulation of spermatogenesis (Bernardino et al., 2019).

**Physiological functions of CA VA and CA VB.** In the urea cycle, the synthesis of citrulline is a rate-limiting step and requires carbamoyl phosphate (CP), which is provided by CPS I. In the mitochondrial matrix, CPS I requires $NH_3$ and $HCO_3^-$ for the synthesis of CP (Cohen, 1981; Lusty, 1978). It has been suggested that the mitochondrial membrane is impermeable to $HCO_3^-$, hence the function of CPS I depends upon mitochondrial CA activity (Vincent & Silverman, 1982). Therefore, mitochondrial CA plays a crucial physiological role in

urea synthesis by making $HCO_3^-$ available to CPS I in the mitochondrial matrix (Vincent & Silverman, 1982). Indeed, the experimental evidence that CA V plays a key role in ureagenesis by providing $HCO_3^-$ to CP I for the synthesis of CP came from a study in which AAZ inhibited the synthesis of citrulline by 71% in guinea-pig liver mitochondria (Dodgson et al., 1983) (Fig. 4).

In addition to its fundamental role in ureagenesis, mitochondrial CA activity is pivotal in gluconeogenesis (Supuran, 2008). The CA inhibitor AAZ has been shown to inhibit gluconeogenesis in both rat and guinea-pig liver (Dodgson & Forster, 1986b; Metcalfe et al., 1985). In the first step of gluconeogenesis, the conversion of pyruvate to oxaloacetate by pyruvate carboxylase (PC) requires $HCO_3^-$, which, in turn, is produced by mitochondrial CA V activity (Dodgson & Forster, 1986a; Häussinger et al., 1986; Hazen et al., 1996). Other studies have shown that pyruvate carboxylation is also required for *de novo* lipogenesis in rat hepatocytes (Lynch et al., 1995; Scozzafava et al., 2013). Taken together, these studies suggest that CA V contributes functionally to several metabolic processes, including at least gluconeogenesis, ureagenesis and *de novo* lipogenesis, by providing $HCO_3^-$ to the intramitochondrial enzymes involved in these metabolic pathways. Subsequently, studies with cultured adipocytes were carried out to investigate the effect of sulfonamide 6-ethoxyzolamide (ETZ) and exogenous bicarbonate/$CO_2$ on pyruvate carboxylation by determining the effect of ETZ on the incorporation of

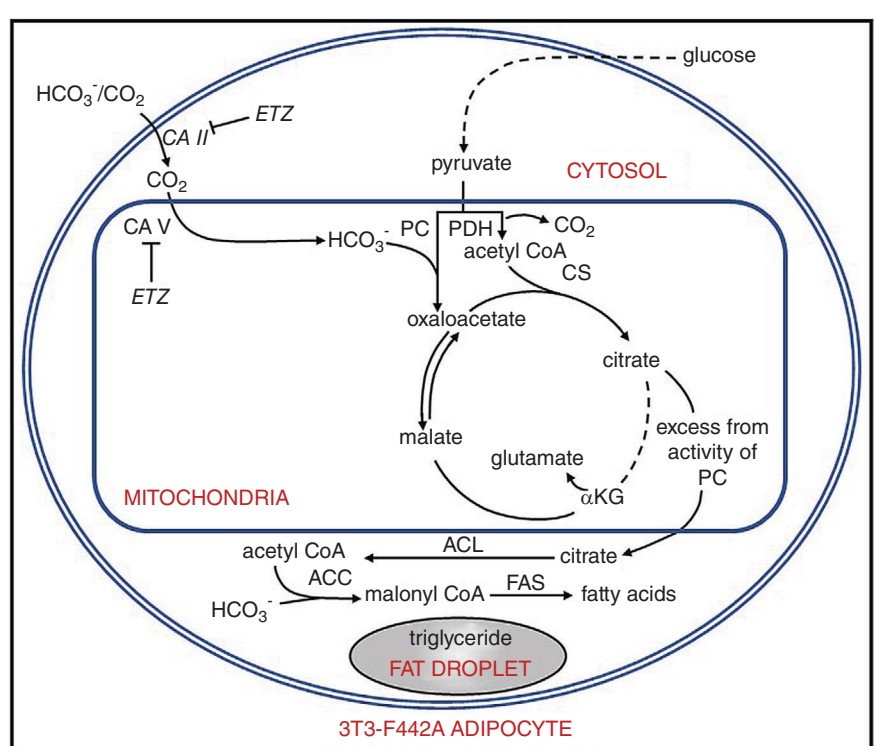

**Figure 4. The physiological role of carbonic anhydrase VA in adipocytes for *de novo* lipogenesis**

Based on the experimental results, it is proposed that the combined activities of the carbonic anhydrases provide enough $HCO_3^-$ for pyruvate carboxylase, which in turn provides citrate for *de novo* lipogenesis and other metabolic intermediates required for other synthetic processes. Abbreviations: ACC, acetyl-CoA carboxylase; ACL, ATP citrate lyase; αKG, α-ketoglutarate; CS, citrate synthase; *ETZ*, 6-ethoxyzolamide; FAS, fatty acid synthetase; PC, pyruvate carboxylase; and PDH, pyruvate dehydrogenase.

[14]C-labelled compounds into weakly acidic metabolites and total lipids and on the concentrations of citrate, malate and ATP. The ETZ caused a $42 \pm 7\%$ decrease in the incorporation of [14]C-labelled bicarbonate into several Krebs cycle intermediates. The concentrations of citrate and malate were also reduced in the presence of ETZ without a concomitant decrease in ATP concentration, suggesting that the decrease in mitochondrial intermediates was not sufficient to inhibit ATP synthesis. It was proposed, however, that it might be sufficient to cause a decrease in the export of mitochondrial citrate to the cytosol. It was further suggested that the decreased export of mitochondrial citrate attributable to CA inhibition by ETZ could explain the significant reduction in *de novo* lipogenesis, because citrate is used by cytosolic ATP citrate lyase to produce acetyl-CoA, the precursor for *de novo* lipogenesis.

In the same study, the presence of mitochondrial CA was examined in adipocytes. Carbonic anhydrase V is present abundantly in adipocytes and plays an important role in maintaining maximal rates of pyruvate carboxylation in these cells (Hazen et al., 1996). It was also discussed that CA II might participate in pyruvate carboxylation by facilitating the diffusion of exogenous $CO_2$ from the cytosol to the alkaline environment of the mitochondrial matrix, where it is converted to $HCO_3^-$ by CA V for use by pyruvate carboxylase (Fig. 4). The findings of this study suggested that the indirect target of ETZ inhibition of *de novo* lipogenesis could be ATP-citrate lyase, owing to a reduction in citrate (Fig. 4). In cells, most citrate is cytosolic, and the reduced citrate concentration decreases ATP-citrate lyase activity and reduces the supply of acetyl-CoA, the precursor for *de novo* lipogenesis (Hazen et al., 1996).

In comparison to CA VA, the physiological role of CA VB is more poorly understood. HuVarBase (Ganesan et al., 2019) introduces six cases of missense mutations of the *CA5B* gene, which have been linked to cancers of the large intestine, stomach, skin and liver. Notably, these mutations might not be causative; only associations have been found. In the *CA5A* gene, HuVarBase reports 15 missense mutations and one deletion. They have been associated with hyperammonaemia and cancers of the large intestine, skin, endometrium and oesophagus. Three of the 15 missense mutations were described as neutral.

The GWAS Central database (https://www.gwascentral.org/index) can be searched to find genome-wide associations between single nucleotide polymorphisms and phenotypes. Using the *P*-value threshold $-\log P \geq 2$, there are two phenotypes significantly associated with markers in the gene or the region of the *CA5A* gene. The phenotype ontology annotation indicates a linkage to Alzheimer's disease and breast cancer. At this *P*-value threshold, the *CA5B* gene shows no phenotypic associations.

**Role of CA VB in human Sertoli cells.** Recently, the role of CAs in hSCs was investigated using expression analysis and CA inhibitors (CAIs), namely, AAZ and ureido-substituted benzenesulfonamide (SLC-0111) (Bernardino et al., 2019). The hSCs expressed three CAs (CA VB, IX and XII), and among them, the expression of CA VB was very high compared with those of the other two CAs. Further analysis suggested that the CAs present in these cells are not involved in the regulation of intracellular pH. Interestingly, hSCs treated with AAZ generated large amounts of lactate and alanine compared with the control group, and CA inhibition altered mitochondrial biogenesis dynamics by regulating the expression of *SIRT1*, *PPARGC1A*, *NRF1* and *HIF1A* (Table 3). *SIRT1* plays a role in cellular energy metabolism and was found to be reduced in the cells treated with AAZ compared with the control group. *PPARGC1A*, which is involved in biogenesis of mitochondria, was downregulated in the cells treated with AAZ. Likewise, there was a significant decrease in the mRNA levels of *NRF1* in AAZ-treated hSCs. Interestingly, when the cells were inhibited using SLC-0111, a CA IX- and XII-specific inhibitor, no changes were observed in the expression levels of these genes except for *PPARGC1A* (Table 3). *HIF1A* mRNA expression was also reduced significantly when treated with AAZ but not in cells treated with SLC-0111 in comparison to control group cells.

Treatment of cells with AAZ for 48 h resulted in a marked reduction in mitochondrial DNA copy number compared with the control group cells. However, there was no change in mitochondrial DNA copy number when the cells were exposed to SLC-0111. In addition, the exposure of the cells to AAZ did not alter any of the mitochondrial complexes (complexes I, II, III, IV and V), suggesting that none of the AAZ-sensitive CAs plays a role in mitochondrial membrane potential in hSCs. Analysis of the effect of CAIs on the accumulation of lipids showed no effect. However, treatment of hSCs with AAZ upregulated hormone-sensitive lipase (*HSL*) mRNA, suggesting a higher rate of lipid degradation owing to CA inhibition. These results demonstrated clearly that inhibition of CA VB, and not CA IX and CA XII, controls the expression of key genes related to mitochondrial biogenesis.

Approximately 40% of seminiferous tubules are filled with Sertoli cells (de França et al., 1993; Sharpe et al., 2003), which represent a key cell type for the maintenance of ion homeodynamics in the seminiferous tubule fluid (Rato et al., 2010). Sertoli cells play an essential role in controlling the maintenance of spermatogenesis by providing metabolic and ionic needs. These cells are responsible for several features of the seminiferous tubule fluid, including ionic composition and pH (Pastor-Soler et al., 2005). The male reproductive tract is known to contain $HCO_3^-$ transporters, which are important for

**Table 3. In Sertoli cells, the mitochondrial carbonic anhydrase VB regulates genes associated with spermatogenesis**

| Gene | Effect of acetazolamide | Effect of SLC-0111* | Suggested role |
|---|---|---|---|
| Hypoxia inducible factor 1 subunit alpha (*HIF1A*) | Decreased levels in the Sertoli cells | No change | Modulates hypoxia responses on gene expression<br>Controls the expression of genes related to mitochondrial biogenesis |
| Sirtuin 1 (*SIRT1*) or NAD-dependent deacetylase sirtuin-1 | Decreased levels in the Sertoli cells | No change | Controls the expression of genes related to mitochondrial biogenesis |
| PPARG coactivator 1 alpha (*PPARGC1A*) | Decreased levels in the Sertoli cells | Decreased levels in the Sertoli cells | Transcriptional coactivator able to upregulate mitochondrial biogenesis, respiratory capacity, oxidative phosphorylation and fatty acid $\beta$-oxidation |
| Nuclear transcription factor 1 (*NRF1*) | Decreased levels in the Sertoli cells | No change | Controls the expression of genes related to mitochondrial biogenesis |
| Hormone-sensitive lipase (*HSL*) | Increased levels in the Sertoli cells | No change | Changes in lipid metabolism |
| Mitochondrially encoded NADH:ubiquinone oxidoreductase core subunit 1 (*MT-ND1*) | Decreased mitochondrial copy number | No change | Controls mitochondrial biogenesis |
| Mitochondrial potential | No change | No change | No role in biogenesis |

*Specific inhibitor of carbonic anhydrase (CA) IX and CA XII (Andreucci et al., 2019).

maintaining the pH in cells. It is also known that $HCO_3^-$ plays an important role in male reproduction (Bernardino et al., 2016). Any change in the concentration of $HCO_3^-$ and the pH in Sertoli cells leads to a disturbance in the ionic balance in the male reproductive tract and can cause infertility (Breton et al., 1996). It has been shown that male mice with *HSL* knocked out are sterile and show increased levels of cholesterol esters in their testes (Osuga et al., 2000). HSL might not play a role in steroidogenesis but might play a different role in reproductive function, because it is believed that the fatty acids released by HSL are required for spermatogenesis. Therefore, the increase in HSL suggests its role in the production of substrates for $\beta$-oxidation of fatty acids in Sertoli cells, and these molecules undergo oxidative phosphorylation via tricarboxylic acid to produce malate. Mitochondrial CAs have been considered targets of CAIs for obesity treatment, because they play a role in lipogenesis (Lynch et al., 1995; Scozzafava et al., 2013). In hSCs, the inhibition of CAs also caused alterations in lipid metabolism, as shown by the increased abundance of HSL in these cells.

To summarize the results on the role of mitochondrial CA in Sertoli cells, it becomes evident that the inhibition of CA VB affects cell metabolism and the expression of genes involved in mitochondrial biogenesis and lipid metabolism. These effects might, in turn, compromise spermatogenesis.

**Inhibition studies on CA VA and CA VB.** Mitochondrial human CA VA is unique owing to its role in several biosynthetic reactions, and it has been shown that CA VA is a potential target for the design of anti-obesity agents (Supuran, 2022). Although it is extremely difficult to design CA isoform-specific inhibitors owing to the very high similarity of the active sites (Alterio et al., 2012; Pinard et al., 2015), it is important to design inhibitors with minimal affinity to other CAs to avoid non-specific effects. Before the design of CA VA- and VB-specific CAIs, the use of sulfonamides, such as the antiepileptic drugs topiramate and zonisamide, resulted in substantial weight loss as a side effect in both human and animal studies (Gordon & Price, 1999; Zareba, 2005). Molecular modelling and X-ray crystallographic studies showed that zonisamide was a more potent inhibitor of CA VA than topiramate (20.6−25.4 nM; De Simone et al., 2005; Vitale et al., 2007). Subsequently, several inhibitors have been synthesized that target CA VA, and their inhibitory activity has been determined. Here, we discuss the

compounds that inhibited the mitochondrial enzymes CA VA and VB most selectively *in vitro* (Figs 5 and 6; Table 4).

The first inhibition study was carried out against murine CA VA by Vullo et al. (2004), using a series of aromatic and heterocyclic sulfonamides. The acylated sulfanilamides and ureido benzenesulfonamides showed higher affinity for mouse CA VA compared with human CA I, human CA II and bovine CA IV. In the second study, a series of aromatic/heterocyclic sulfonamides containing phenacetyl, pyridylacetyl and thienylacetyl tails were studied, and these compounds selectively inhibited

human CA VA and human CA VB over CA I and CA II (Güzel et al., 2009). In addition, an inhibition study focusing on human CA VB showed that some compounds, especially sulpiride, selectively inhibited CA VB compared with CA II (Nishimori et al., 2005). An investigation using 4-(4-phenyltriazole-1-yl)-benzenesulfonamide derivatives as inhibitors of CA VA and CA VB found that many of them selectively inhibited CA VA or CA VB (Poulsen et al., 2008). An assay using a series of aromatic/heterocyclic sulfonamides containing fructopyranose-thioureido tails found a single compound (a metanilamide derivative) that

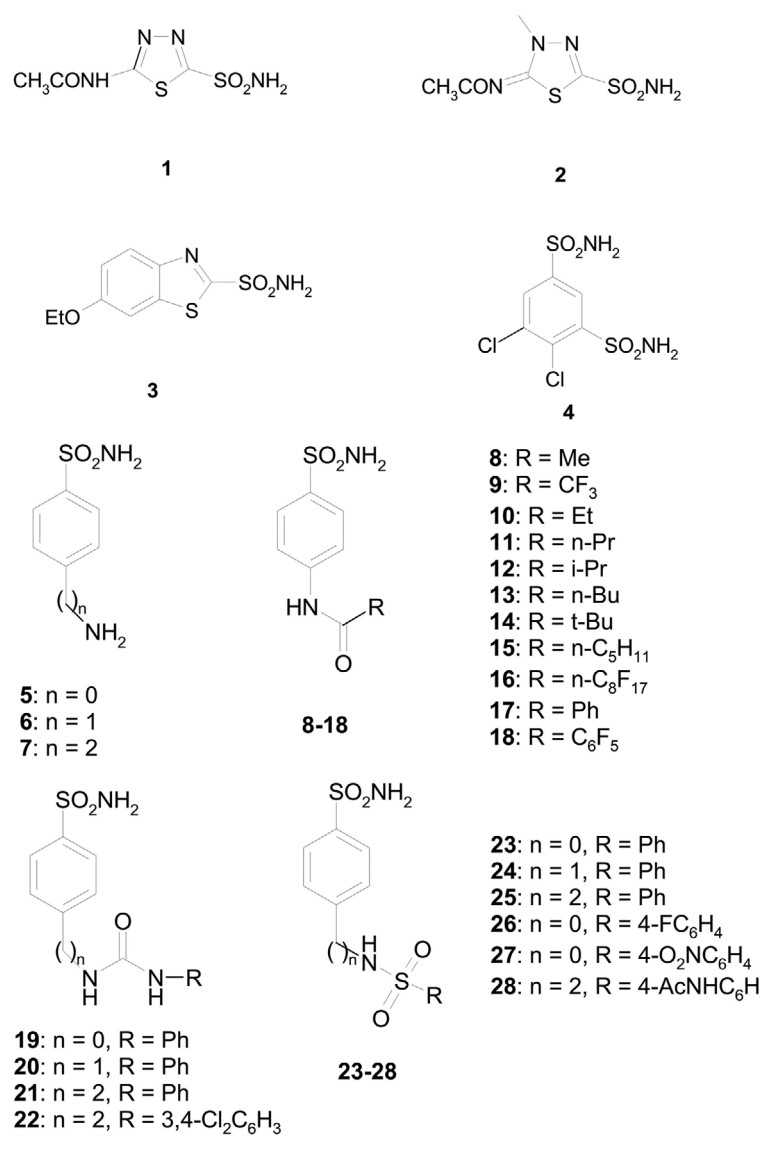

**Figure 5. Structures of the inhibitor molecules**
Chemical structures of aromatic/heterocyclic sulfonamide carbonic anhydrase inhibitors (CAIs) 1–29 tested as carbonic anhydrase VA/VB inhibitors, which led to the observation of low nanomolar inhibition constants (Smaine et al., 2008; Vullo et al., 2004).

**Table 4. Inhibition studies of carbonic anhydrase VA and VB using classical and non-classical inhibitors**

| CA inhibitors | | Human* or mouse† | | |
|---|---|---|---|---|
| Class | Derivatives | CA VA (nM) | CA VB (nM) | Reference |
| Sulfonamides | Acylated sulfanilamides and the ureido benzenesulfonamides 1,3,4-thiadiazole-2-sulfonamide and aminobenzolamide | 4–15 | — | Vullo et al. (2004)† |
| | *N*-(2-fluoro-4-sulfamoyl-phenyl)-2-(2-thienyl) acetamide and *N*-(2-bromo-4-sulfamoyl-phenyl)-2-phenylacetamide | 5.9–10.2 | 5.9–10.2 | Güzel et al. (2009)* |
| | 4-(4-Phenyltriazole-1-yl)-benzenesulfonamide | 19.6–9.3 | 10.5–54.2 | Poulsen et al. (2008)* |
| | 2-Substituted-1,3,4-thiadiazole-5-sulfamides | 4.2–32 | 1.3–74 | Smaine et al. (2008)* |
| | Thiadiazolesulfonamide | — | — | Maresca & Supuran (2011)* |
| | Difluoromethanesulfonamides | 160 | — | Cecchi et al. (2005)* |
| | 5-[2-(Benzimidazol-1-yl) acetyl]-2-chloro-benzenesulfonamide and its analogues | 0.25, 0.77 and 1.82 | 22.2 | Čapkauskaitė et al. (2018)* |
| Phenols | 2-(4-Hydroxyphenyl) acetamide | 70–125 | 70–125 | Davis et al. (2010)* |
| Coumarin | Metronidazole–coumarin conjugates and 3-cyano-7-hydroxycoumarin | 0.38–2.63 $\mu$M | 0.38–2.63 $\mu$M | Bonneau et al. (2013)* |

Abbreviation: CA, carbonic anhydrase.

distinguished mitochondrial CA VA from CA II (Winum et al., 2007). A study involving a small series of 2-substituted-1,3,4-thiadiazole-5-sulfamides showed compounds with selectivity for CA VA and CA VB

**Figure 6. Structures of compounds that inhibit carbonic anhydrase VA/VB efficiently**
Sulfonamides with low nanomolar carbonic anhydrase VA/VB inhibitory action reported by Güzel et al. (2009). In compounds 32 and 33, $X_1$ and $X_2$ are either CH or N.

compared with CA I, CA II and CA IV, with a selectivity ratio for inhibiting CA VA and CA VB over CA II in the range of 67.5–415, making these sulfamides promising inhibitors of CA VA and VB (Smaine et al., 2008). Maresca & Supuran (2011) found that among the series of (*R*)-/(*S*)-10-camphorsulfonyl-substituted aromatic/heterocyclic compounds, thiadiazolesulfona-mide was the most efficient inhibitor that selectively inhibited CA VA and VB compared with CA I and II. Among the several derivatives of aromatic/heteroaromatic/polycyclic difluoromethanesul-fonamides assayed against human CAs, some of them, such as 3-bromophenyl-difluoromethanesul-fonamide, showed selectivity for CA VA over CA I, II and IX (Cecchi et al., 2005). Recent studies using *N*-alkylated benzimidazoles identified several compounds that bind selectively to human CA VA; among them, 5-[2-(benzimidazol-1-yl)acetyl]-2-chloro-benzenesulfo-namide and its analogues showed high affinity for both CA VA and CA VB compared with the other CA isoforms tested (Čapkauskaitė et al., 2018).

Inhibition studies using natural product-based phenolic compounds against CA I, II, VA and VB showed selectivity for CA VA and VB over CA I and II, with selectivity ratios of 120–3800. The study identified 2-(4-hydroxyphenyl)acetamide as one of the best CA VA- and CA VB-selective inhibitors (Davis et al., 2010). The selectivity ratios for inhibiting the mitochondrial over the cytosolic isoforms for these phenol derivatives were in the range of 120–3800,

making them the most isoform-selective compounds for inhibiting human CA VA/VB known to date. Studies using coumarin derivatives incorporating a nitroazole moiety and 3-cyano-7-hydroxy-coumarin showed significant inhibition of the mitochondrial isoforms CA VA and VB compared with CA I, II, IV and XIII (Bonneau et al., 2013). Given that CA VA and VB are involved in several biosynthetic processes catalysed by pyruvate carboxylase, acetyl-CoA carboxylase and CPS I and II, providing the bicarbonate substrate to these carboxylating enzymes involved in fatty acid biosynthesis, these inhibitors were considered promising leads for the development of anti-obesity agents with a novel mechanism of action.

**Loss-of-function mutations in CA VA lead to hyperammonaemia.** Inborn errors of metabolism are rare genetic disorders resulting from defects in a metabolic enzyme owing to a mutation in a single gene. Defective enzymes in biochemical and metabolic pathways can affect the metabolism of proteins, fats or carbohydrates, leading to complicated medical conditions involving several human organ systems. Among the inborn errors of metabolism, hyperammonaemia is a genetic disorder that requires immediate treatment. The aetiology of hyperammonaemia is heterogeneous; it can be a result of to genetic, developmental or environmental factors (Häberle, 2013). In the recent past, two studies presented children from several families with hyperammonaemia, and the underlying cause in each affected child was the deficiency of CA VA (CA5A [MIM 114671]) (Diez-Fernandez et al., 2016; van Karnebeek et al., 2014).

In the first study, four children from unrelated families showed hyperammonaemic encephalopathy and hyperlactataemia. The first family consisted of healthy non-consanguineous parents of Belgian–Scottish descent and their three children. Among them, the affected sister and brother developed lethargy, tachypnoea, hypoglycaemia, hyperlactataemia, hypernatraemia and hyperammonaemia with respiratory alkalosis within the first days of life. Analyses of organic acids from urine showed

excretion of higher levels of lactic, $\beta$-hydroxybutyric and acetoacetic acids, with increases in carboxylase substrates and related metabolites. In addition, the children showed elevated levels of glutamine, alanine and proline and reductions in citrulline and arginine in the plasma, with no defects other other than the mutation in the *CA5A* gene (van Karnebeek et al., 2014). Behavioural studies showed below average motor coordination. The affected children were homozygous, with a single nucleotide change in the *CA5A* gene that led to a change in the protein, with a substitution at position 233 (Ser to Pro), which disrupted the structure around the conserved Thr235 residue that forms part of the substrate-binding region of the enzyme (Fig. 7). The Ser233 residue is highly conserved across human CA isoforms, and studies have shown that mutations in this region make the structure unstable around the substrate-binding region, thus reducing the activity of the enzyme (Krebs & Fierke, 1993). *In vitro* mutation studies (Ser233Pro) showed a reduction in the activity of the mutant protein to 20% at 30°C compared with the wild-type protein. Incubation at 37°C retained only 5% of its activity, suggesting that the mutant protein is sensitive to temperature. Interestingly, administration of dextrose, bicarbonate and enteral carglumic acid (Carbaglu) normalized the clinical and metabolic findings in the children.

In the second family, a male child born to non-consanguineous Russian parents presented with lethargy, weight loss, jaundice and tachypnoea on day 4 of life. In addition, the child showed hyperammonaemia, hyperlactataemia mild hypoglycaemia, metabolic acidosis and ketonuria. Sequencing of *CA5A* exons revealed a synonymous c.555G>A change at the final base of exon 4 (Fig. 7). Further analyses of the exons showed that the mutated *CA5A* generated a product of a different size that led to an in-frame deletion of exon 4 from the RNA (Krawczak et al., 2007). Comparison of the truncated RNA with the other CAs identified three residues (His155, Tyr164 and Tyr167) from the deleted *CA5A* transcript (residues 154—185) that are part of the active site of the enzyme (Nagao et al., 1993). Thus, the deletion was predicted to reduce the activity of the enzyme, or it

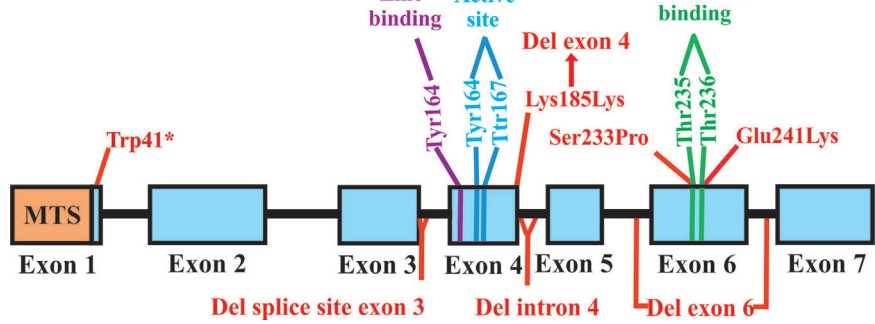

**Figure 7. Mapping of *CA5A* mutations affecting the function of the protein**
The presented human wild-type *CA5A* gene showing the predicted mitochondrial targeting sequence (MTS) and the functional residues for zinc binding (purple), the active site (blue) and substrate binding (green). The mutations affecting protein function are shown in red.

**Table 5. Carbonic acid VA deficiency in humans downregulates important enzymes of the mitochondrial matrix**

| Enzyme deficiency | Increased biochemical parameters | Decreased biochemical parameters | References |
|---|---|---|---|
| Carbamoyl phosphate synthetase | Ammonia and glutamine | Citrulline and arginine | Diez-Fernandez et al. (2016); van Karnebeek et al. (2014) |
| Pyruvate carboxylase | Redox imbalance (lactate and dicarboxylic acids) and lysine | Gluconeogenesis and tricyclic acid cycle intermediates (cataplerosis) | Diez-Fernandez et al. (2016); van Karnebeek et al. (2014) |
| Proprionyl-CoA carboxylase | 3-OH-propionic acid, propionylglycine and methylcitrate | — | van Karnebeek et al. (2014) |
| 3-Methylcrotonyl-CoA carboxylase | 3-Methylcrotonylglycine and 3-OH-isovaleric acid | — | van Karnebeek et al. (2014) |

might also lead to misfolding and degradation of the affected protein. The administration of carglumic acid and biotin along with protein-free formula and intravenous lipids was shown to resolve the metabolic acidosis and hyperammonaemia.

In the third family, a male child born to first-cousin consanguineous Pakistani parents was encephalopathic, with hyperammonaemia and hyperlactataemia and with compensated metabolic acidosis at the age of 13 months. Sequencing of exons revealed a homozygous deletion of the 4 kb region that included exon 6 in the two siblings of the family that led to the absence of CA VA protein, as confirmed by immunoblotting.

The observations of the first study, that hyperammonaemia is amenable to treatment and that it is thus possible to prevent irreversible brain damage, motivated the second study to investigate further the effect of defective hepatic $HCO_3^-$ production as the cause of hyperammonaemic encephalopathy (Diez-Fernandez et al., 2016). In this study, children showed hyperammonaemia, in whom the biochemical profile was not indicative of any other urea cycle disorder. Sequencing of the *CA5A* and *CA5B* genes was performed in 96 children who were considered to suffer from a proximal urea cycle disorder but had no mutations in *N*-acetylglutamate synthase (*NAGS*) and *CPS1*. Among these children, *CA5A* mutations were found in 10 patients who showed a unique combination of biochemical findings, including hyperammonaemia, elevated lactate, and elevated ketone bodies in urine (Table 5). Metabolic acidosis and urinary excretion of carboxylase substrates and related metabolites were also observed in variable ranges. Seven of the 10 newly described hyperammonaemic children presented with only one initial hyperammonaemic crisis, after which they remained stable, even after tapering of treatment. However, three newly described patients suffered a second crisis, which was milder than the first one.

Mutational analyses of these children showed a splice-site deletion of exon 3 in one patient that resulted in the skipping of exon 3, leading to a frame shift, p.(Gly114Ala fs*53) (Fig. 7). The remaining six children showed an in-frame deletion of either exon 4 or exon 6, causing inactivation of the enzyme or misfolding and degradation of the protein (Fig. 7 and Table 6).

Characterization of CA VA recombinant enzymes of wild-type, mutant and the three non-disease-associated variants (p. Asn45Lys, p. Asn46Lys, and p. Pro237Leu) showed decreased production of mutant proteins (exons 4 and 5 deleted) and reduced thermal stability at ∼4°C compared with wild-type CA VA (Diez-Fernandez et al., 2016). Deletions of exons 4 and 6 reduced the stability of the protein, making the enzyme inactive and therefore causing hyperammonaemia in the children, because exon 4 encodes both the zinc-binding and active sites of the enzyme, and exon 6 contains substrate-binding sites. The two mutant recombinant proteins with missense mutations (p. Ser233Pro, p. Glu241Lys and p. Glu241Lys) showed decreased (50–75%) catalytic activity, suggesting that the mutated residues affect the binding of the substrate in the substrate-binding site. In addition to the decreased catalytic activity, the CA VA variants of p. Ser233Pro, p. Glu241Lys showed decreased thermal stability that might result in shorter half-lives of these two mutant forms in liver mitochondria.

Detailed analyses of the phenotypes of CA VA-deficient children showed unique biochemical findings during the neonatal period (van Karnebeek et al., 2014). Interestingly, although the biochemical findings were not all consistently abnormal in all the children described in these studies, hyperammonaemia, hyperlactataemia and ketonuria were reported in all cases. The findings were consistent with the impairment of all four enzymes that require $HCO_3^-$ generated by CA VA in the liver mitochondria (Table 5 and Fig. 8).

**Table 6. Details of the patients in two studies with a mutation in *CA5A* leading to carbonic acid VA deficiency**

| No. | Mutation in *CA5A** | Amino acid change[†] | Neurological outcome | Ethnicity | Parental relationship | References |
|---|---|---|---|---|---|---|
| 1 | **Exon 1** c.123G>A | p.Trp41 Truncated protein | Normal at 12 months | Turkish | Consanguineous | Diez-Fernandez et al. (2016) |
| 2 | **Exon 3** c.458_459+ 22del24 bp | p.? Predicted splicing | Normal at 10 years | Indian | Consanguineous (not first cousins) | Diez-Fernandez et al. (2016) |
| 3 | **Exon 4** c.555G>A | p.His155_Leu186del Deletion of exon 4 | Normal at 6 months | Russian | Non-consanguineous | van Karnebeek et al. (2014) |
| 4 | **Intron 4** c.555+4_555+ 183del180 bp | p.? Predicted splicing | Normal at 10 years | Pakistani | Consanguineous (not first cousins) | Diez-Fernandez et al. (2016) |
| 5 | **Exon 6** c.697T>C | p.ser233Pro | Normal at 4.5 years; below-average motor coordination | Belgian– Scottish | Non-consanguineous | van Karnebeek et al. (2014) |
| 6 | **Exon 6** c.697T>C | p.ser233Pro | Development below average | Belgian– Scottish | Non-consanguineous | van Karnebeek et al. (2014) |
| 7 | **Exon 6** c.721G>A | p.Glu241Lys | Normal | Bangladeshi | Consanguineous | Diez-Fernandez et al. (2016) |
| 8 | **Exon 6** c.721G>A | p.Glu241Lys | Normal; no treatment at 3 years | Pakistani | Not reported | Diez-Fernandez et al. (2016) |
| 9 | **Exon 6** c.619-3420_c.774+ 502del4078 bp | p.(Asp207_Gln258del) Exon 6 del | Normal at 11 years | Pakistani | First-cousin consanguineous | van Karnebeek et al. (2014) |
| 10 | **Exon 6** c.619-3420_c.774+ 502del4078 bp | p.(Asp207_Gln258del) Exon 6 del | Normal; no treatment at 6 years | Pakistani | First-cousin consanguineous | van Karnebeek et al. (2014) |
| 11 | **Exon 6** c.619-3420_c.774+ 502del4078 bp | p.(Asp207_Gln258del) Exon 6 del | Normal at 4 years | Indian | Consanguineous (not first cousins) | van Karnebeek et al. (2014) |
| 12 | **Exon 6** c.619-3420_c.774+ 502del4078 bp | p.(Asp207_Gln258del) Exon 6 del | Normal; no treatment at 4 years | Pakistani | Not reported | van Karnebeek et al. (2014) |
| 13 | **Exon 6** c.619-3420_c.774+ 502del4078 bp | p.(Asp207_Gln258del) Exon 6 del | Learning difficulties and speech delay at 5 years | Pakistani | First-cousin consanguineous | van Karnebeek et al. (2014) |
| 14 | **Exon 6** c.619-3420_ c.774+502del4078 bp | p.(Asp207_Gln258del) Exon 6 del | Normal; no treatment at 9 months | Pakistani | Non-consanguineous | van Karnebeek et al. (2014) |

*The upper line in bold shows exons/introns affected in the children. The reference sequence for *CA5A* was Ensembl ENSG00000174990. Mutations were found in all patients in a homozygous state. The result of the mutation at the protein level is indicated in the lower line (except for missense mutations).
[†]Translation of the complementary DNA GenBank RefSeq NM_0 01739.1 and nucleotide 57 in this sequence is considered +1 because it is the A of the translation initiation codon.

In these studies, 65% of the children with CA VA defects remained stable after the initial crisis, and the remaining 35% experienced another crisis that was less severe than the initial crisis. This might be attributable to the presence of CA VB that is not sufficiently available at the time of birth but later becomes functional, thus preventing a severe metabolic crisis later in life (van Karnebeek et al., 2014).

Similar to humans, the mouse knockout model for *Car5A* presented with life-threatening hyperammonaemia. In contrast, *Car5B* knockout mice showed no clinical phenotype. Interestingly, the double-knockout mice for both *Car5A* and *Car5B* developed more severe hyperammonaemia than the *Car5A* knockout mice, suggesting that although both CA VA and VB seem to contribute to ureagenesis in mice, CA VA has a predominant role (Shah et al., 2013).

Carbonic anhydrase VA deficiency in humans seems to be more common than other rare metabolic diseases,

and early identification might allow specific treatment of hyperammonaemia and, ultimately, prevent neurological sequelae (Summar et al., 2013).

**Conclusions.** Among the mammalian CAs, CA VA was one of the first enzymes identified to play crucial roles in ureagenesis, gluconeogenesis and lipogenesis. The mammalian CA VA and CA VB are the only two enzymes that are expressed in the mitochondria and have very interesting patterns of expression. Carbonic anhydrase VA is predominantly expressed in the liver, and CA VB is present in many tissues except the liver. The reasons for such interesting patterns of expression are not well understood.

Carbonic anhydrase VA has been of particular interest to understand the precise role of this enzyme and to design CA-specific inhibitor compounds to target the enzyme, whereas much less is known about CA VB. Inhibition of CA VA using AAZ, a carbonic anhydrase

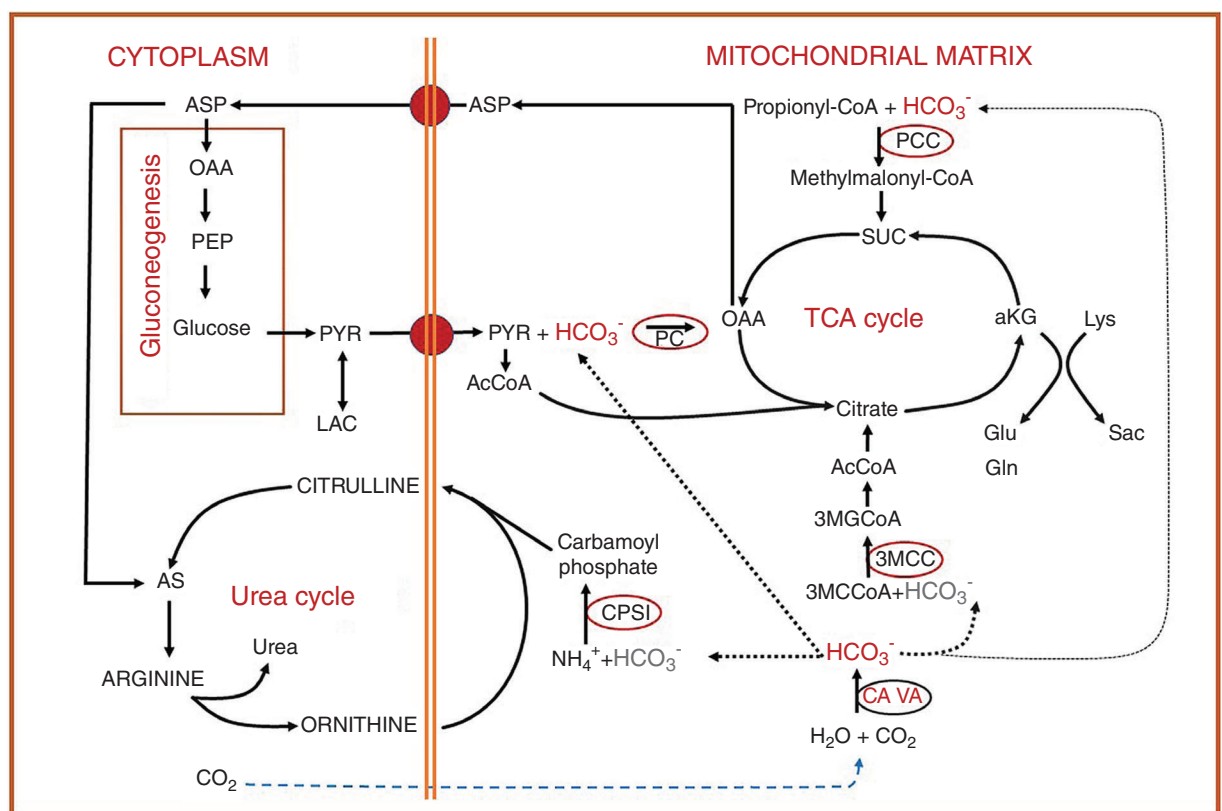

**Figure 8. Effects of carbonic anhydrase VA deficiency on biochemical pathways in humans**
The dashed lines show the supply of $HCO_3^-$ produced by carbonic anhydrase VA to bicarbonate-dependent carboxylases. The enzyme affected owing to a lack of $HCO_3^-$ is indicated with red circles. Enzymes are as follows: CPS I, carbamoylphosphate synthetase I; PC, pyruvate carboxylase; PCC, propionyl CoA carboxylase; and 3MCC, 3-methylcrotonyl CoA carboxylase. Metabolites are as follows: AcCoA, acetyl-CoA; $\alpha$KG, $\alpha$-ketoglutarate; AS, arginino-succinate; ASP, aspartate; Gln, glutamine; Glu, glutamate; LAC, lactate; Lys, lysine; 3MCCoA, 3-methylcrotonyl CoA; 3MGcCoA, 3-methylglutaconyl CoA; OAA, oxaloacetate; PEP, phosphoenolpyruvate; PYR, pyruvate; Sac, saccharopine; and SUC, succinyl CoA.

inhibitor, revealed that the main function of CA VA is to provide $HCO_3^-$ ions to the enzymes in mitochondria that are involved in several important biochemical pathways. Subsequent studies using a knockout mouse model confirmed the role of CA VA in detoxification of ammonia (ureagenesis) through CPS I in mitochondria. In addition, recent studies involving children have shown that deficiency of CA VA owing to mutation in the *CA5A* gene leads to hyperammonaemia, providing further confirmation of its role in ureagenesis.

The precise physiological role of CA VB, another mitochondrial enzyme, is not known. Interestingly, no human mutations have been found in *CA5B* until now. Although *Car5B* null mice showed no major abnormalities, its role in ureagenesis was evident only in the absence of *Car5A*. In addition, knockout studies reported that *Car5A* null mice showed poor growth and that mice null for both *Car5A* and *Car5B* had lower fertility. In addition, the survival of the male mice was reduced after weaning. Recent inhibition studies involving Sertoli cells have suggested that CA VB plays an important role in spermatogenesis, although no reproductive phenotype was observed in mice deficient only in CA VB.

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

## Additional information

### Competing interests

None.

### Author contributions

A.A. and S.P. are responsible for conceiving the initial idea and for preparing the first version of the manuscript. All authors contributed to the writing and revision of the article. All authors have read and approved the manuscript and agree to be accountable for all aspects of the work in ensuring that questions related to the accuracy or integrity of any part of the work are appropriately investigated and resolved. All persons designated as authors qualify for authorship, and all those who qualify for authorship are listed.

### Funding

None.

### Acknowledgements

We are grateful to the Finnish Cultural Foundation, Tampere Tuberculosis Foundation, Jane & Aatos Erkko Foundation, Finnish Foundation for Cardiovascular Research and the Academy of Finland.

### Keywords

bicarbonate ion, carbamoylphosphate synthetase I, carbonic anhydrase VA, carbonic anhydrase VB, hyperammonaemia, mitochondria

## Supporting information

Additional supporting information can be found online in the Supporting Information section at the end of the HTML view of the article. Supporting information files available:

**Peer Review History**

