## [Peer Review History · The Journal of Physiology]

Mitochondrial carbonic anhydrase VA and VB: Properties and roles in health and disease

Ashok Aspatwar, Claudiu T. Supuran, Abdul Waheed, William S Sly, and Seppo Parkkila
DOI: 10.1113/JP283579

Corresponding author(s): Seppo Parkkila (seppo.parkkila@tuni.fi)

Review Timeline:

Submission Date:	17-Aug-2022
Editorial Decision:	25-Oct-2022
Revision Received:	20-Nov-2022
Accepted:	30-Nov-2022

Senior Editor: Laura Bennet

Reviewing Editor: Kyle McCommis

Transaction Report:

Dear Professor Parkkila,

Re: JP-TR-2022-283579 "Mitochondrial carbonic anhydrase VA and VB: Properties and roles in health and disease" by Ashok Aspatwar, Claudiu T. Supuran, Abdul Waheed, William S Sly, and Seppo Parkkila

Thank you for submitting your Topical Review to The Journal of Physiology. It has been assessed by a Reviewing Editor and by 2 expert referees and I am pleased to tell you that it is considered to be acceptable for publication following satisfactory revision.

The reports are copied at the end of this email. Please address all of the points and incorporate all requested revisions, or explain in your Response to Referees why a change has not been made.

NEW POLICY: In order to improve the transparency of its peer review process The Journal of Physiology publishes online as supporting information the peer review history of all articles accepted for publication. Readers will have access to decision letters, including all Editors' comments and referee reports, for each version of the manuscript and any author responses to peer review comments. Referees can decide whether or not they wish to be named on the peer review history document.

I hope you will find the comments helpful and have no difficulty in revising your manuscript within 4 weeks.

Your revised manuscript should be submitted online using the links in Author Tasks Link Not Available. This link is to the Corresponding Author's own account, if this will cause any problems when submitting the revised version please contact us.

You should upload:

- A Word file of the complete text (including any Tables);
- An Abstract Figure, (with accompanying Legend in the article file)
- Each figure as a separate, high quality, file;
- A full Response to Referees;
- A copy of the manuscript with the changes highlighted.
- Author profile. A short biography (no more than 100 words for one author or 150 words in total for two authors) and a portrait photograph of the two leading authors on the paper. These should be uploaded, clearly labelled, with the manuscript submission. Any standard image format for the photograph is acceptable, but the resolution should be at least 300 dpi and preferably more.

- A 'Cover Art' file for consideration as the Issue's cover image;
- Appropriate Supporting Information (Video, audio or data set https://jp.msubmit.net/cgi-bin/main.plex?form_type=display_requirements#supp).

To create your 'Response to Referees' copy all the reports, including any comments from the Senior and Reviewing Editors into a Word, or similar, file and respond to each point in colour or CAPITALS. Upload this when you submit your revision.

I look forward to receiving your revised submission.

Yours sincerely,

Professor Laura Bennet
Senior Editor
The Journal of Physiology
<https://jp.msubmit.net>
<http://jp.physoc.org>
The Physiological Society
Hodgkin Huxley House
30 Farringdon Lane
London, EC1R 3AW
UK
<http://www.physoc.org>
<http://journals.physoc.org>

EDITOR COMMENTS

Reviewing Editor:

Thank you for submitting your review manuscript titled "Mitochondrial carbonic anhydrase VA and VB: Properties and roles in health and disease" to The Journal of Physiology. We must apologize for the lengthy delay, it required a substantial amount of time to secure 2 independent reviewers. This manuscript has now been assessed by 2 academic peer reviewers and an academic reviewing editor. I am pleased to say that all reviewers believe this review article to be very well written, and likely to have high influence on the field of metabolic physiology and rare diseases. Both reviewers identified minor issues in the writing and have several minor critiques which should be easy to address. We look forward to receiving your revised review article.

Please see 'required items' below.

REFeree COMMENTS

Referee #1:

This is a well-written, concise and up-to-date review on the two mitochondrial carbonic anhydrase isoforms VA and VB, respectively. Physiology, pathophysiology, pharmacology and role in human disease are discussed.

I only have minor comments that may help to improve the manuscript:

1. Figures 4 and 8 could be improved in quality, design and color choice. Furthermore, the font size should be increased.
2. The authors repeatedly mention the putative role of CA VB in male spermatogenesis and fertility, statements which are largely based on in vitro studies. But CA VB KO mice have been developed and characterized - some of the authors have been involved in these studies. No apparent phenotype of CA VB Ko mice was reported. Is there evidence that male CA VB KO mice exhibit reduced fertility ? If this is not the case, then this should be mentioned in the text.
3. GWAS: have any phenotypic traits been linked to SNPs in the region of the CA VA or CA VB genes ?

Referee #2:

The paper by Aspatwar et al. entitled "Mitochondrial carbonic anhydrase VA and VB: Properties and roles in health and disease" is an interesting review regarding these two mentioned mitochondrial isoforms. The review is very well written and dissects all the aspects of these two isoforms, from the biochemical to the functional features and the inhibition studies carried out on the two enzymes.

The presented review gives complete information on recently progress made in the field.

Only some minor revisions are necessary:

- 1) Abstract figure: Rephrase last two lines regarding presence of CA VA and B, specify which are the organs where CA VB is present.
- 2) Pag 4, line 14, an "and" is missing between VA VB.
- 3) Pag 5, Biochemistry CA VA and CA VB. It would be clearer to the reader if authors would insert a table regarding the catalytic efficiency of these two isoforms compared to the other isoforms, rather than listing them in the text.
- 4) Pag 6 and 7, Authors repeat twice the information regarding the amino acidic similarity of CA VB to CA VA (66%). Please edit.
- 5) Pag 7, Authors should include in the text the information that when protein numbering is not inclusive of the signal peptide, residue position is annotated with a + before the residue number. This can help the reader and avoid confusion when numbered residues are cited within the manuscript.

6) Pag 7, CA VB contains 6 cysteines and not 5 as mentioned in the text. Please correct the sentences in the text.

7) Pag 8, Fig 3, please box the cysteine in position 92 of CA VA and VB, and edit Figure legend.

8) Pag 7 Authors state that CA VB may contain four Cys residues possibly involved in intradisulfide bond formation referring to reference 39 which is a PNAS manuscript on CA IV. Please note that in CA IV the cysteines involved into disulfide bonds are not conserved in CA VB. Please check the reference. Moreover, the crystallographic structure of murine CA V was solved showing that all the cysteine residues are reduced. Why Authors do not use this structure to predict the CA VB redox state of cysteines? Please check

REQUIRED ITEMS:

-Your MS must include a complete "Additional information section" with the following 4 headings and content:

Competing Interests: A statement regarding competing interests. If there are no competing interests, a statement to this effect must be included. All authors should disclose any conflict of interest in accordance with journal policy.

Author contributions: Each author should take responsibility for a particular section of the study and have contributed to writing the paper. Acquisition of funding, administrative support or the collection of data alone does not justify authorship; these contributions to the study should be listed in the Acknowledgements. Additional information such as 'X and Y have contributed equally to this work' may be added as a footnote on the title page.

It must be stated that all authors approved the final version of the manuscript and that all persons designated as authors qualify for authorship, and all those who qualify for authorship are listed.

Funding: Authors must indicate all sources of funding, including grant numbers. If authors have not received funding, this must be stated.

It is the responsibility of authors funded by RCUK to adhere to their policy regarding funding sources and underlying research material. The policy requires funding information to be included within the acknowledgement section of a paper. Guidance on how to acknowledge funding information is provided by the Research Information Network. The policy also requires all research papers, if applicable, to include a statement on how any underlying research materials, such as data, samples or models, can be accessed. However, the policy does not require that the data must be made open. If there are considered to be good or compelling reasons to protect access to the data, for example commercial confidentiality or legitimate sensitivities around data derived from potentially identifiable human participants, these should be included in the statement.

Acknowledgements: Acknowledgements should be the minimum consistent with courtesy. The wording of acknowledgements of scientific assistance or advice must have been seen and approved by the persons concerned. This section should not include details of funding.

-The Reference List must be in Journal format https://jp.msubmit.net/cgi-bin/main.plex?form_type=display_requirements#refs

-Please upload separate high quality figure files via the submission form.

-It is the authors' responsibility to obtain any necessary permissions to reproduce previously published material https://jp.msubmit.net/cgi-bin/main.plex?form_type=display_requirements#use

END OF COMMENTS

Confidential Review

17-Aug-2022

The paper by Aspatwar et al. entitled "Mitochondrial carbonic anhydrase VA and VB: Properties and roles in health and disease" is an interesting review regarding these two mentioned mitochondrial isoforms. The review is very well written and dissects all the aspects of these two isoforms, from the biochemical to the functional features and the inhibition studies carried out on the two enzymes.

The presented review gives complete information on recently progress made in the field.

Only some minor revisions are necessary:

- 1) Abstract figure: Rephrase last two lines regarding presence of CA VA and B, specify which are the organs where CA VB is present.
- 2) Pag 4, line 14, an "and" is missing between VA VB.
- 3) Pag 5, Biochemistry CA VA and CA VB. It would be clearer to the reader if authors would insert a table regarding the catalytic efficiency of these two isoforms compared to the other isoforms, rather than listing them in the text.
- 4) Pag 6 and 7, Authors repeat twice the information regarding the amino acidic similarity of CA VB to CA VA (66%). Please edit.
- 5) Pag 7, Authors should include in the text the information that when protein numbering is not inclusive of the signal peptide, residue position is annotated with a + before the residue number. This can help the reader and avoid confusion when numbered residues are cited within the manuscript.
- 6) Pag 7, CA VB contains 6 cysteines and not 5 as mentioned in the text. Please correct the sentences in the text.
- 7) Pag 8, Fig 3, please box the cysteine in position 92 of CA VA and VB, and edit Figure legend.
- 8) Pag 7 Authors state that CA VB may contain four Cys residues possibly involved in intradisulfide bond formation referring to reference 39 which is a PNAS manuscript on CA IV. Please note that in CA IV the cysteines involved into disulfide bonds are not conserved in CA VB. Please check the reference. Moreover, the crystallographic structure of murine CA V was solved showing that all the cysteine residues are reduced. Why Authors do not use this structure to predict the CA VB redox state of cysteines? Please check

Professor Laura Bennet
Senior Editor
The Journal of Physiology
The Physiological Society
Hodgkin Huxley House
30 Farringdon Lane
London, EC1R 3AW
UK

November 14th, 2022

Dear professor Bennet,

We were grateful for the constructive critiques our manuscript (JP-TR-2022-283579 "Mitochondrial carbonic anhydrase VA and VB: Properties and roles in health and disease") received and feel that we were able to respond to the questions or suggestions with changes that greatly improved our work. Below you can find the modifications made according to reviewers' comments. We also corrected some typos and other errors found in the text during the editing process. I hope that you will find the revised version of the manuscript acceptable for publication in the Journal of Physiology.

Sincerely yours,

Seppo Parkkila, professor, dean
Faculty of Medicine and Health Technology
Tampere University, Finland

Reviewing Editor:

Thank you for submitting your review manuscript titled "Mitochondrial carbonic anhydrase VA and VB: Properties and roles in health and disease" to The Journal of Physiology. We must apologize for the lengthy delay, it required a substantial amount of time to secure 2 independent reviewers. This manuscript has now been assessed by 2 academic peer reviewers and an academic reviewing editor. I am pleased to say that all reviewers believe this review article to be very well written, and likely to have high influence on the field of metabolic physiology and rare diseases. Both reviewers identified minor issues in the writing and have several minor critiques which should be easy to address. We look forward to receiving your revised review article.

- **We thank the Reviewing Editor for these very positive comments on our manuscript.**

Reviewer #1

This is a well-written, concise and up-to-date review on the two mitochondrial carbonic anhydrase isoforms VA and VB, respectively. Physiology, pathophysiology, pharmacology and role in human disease are discussed.

I only have minor comments that may help to improve the manuscript:

1. Figures 4 and 8 could be improved in quality, design and color choice. Furthermore, the font size should be increased.

Our response: The new versions of Figures 4 and 8 have been improved for better clarity.

2. The authors repeatedly mention the putative role of CA VB in male spermatogenesis and fertility, statements which are largely based on in vitro studies. But CA VB KO mice have been developed and characterized - some of the authors have been involved in these studies. No apparent phenotype of CA VB Ko mice was reported. Is there evidence that male CA VB KO mice exhibit reduced fertility? If this is not the case, then this should be mentioned in the text.

Our response: We have modified the last sentence of Conclusions as follows: "Recent inhibition studies involving Sertoli cells suggested that CA VB plays an important role in spermatogenesis, although no reproductive phenotype was observed in mice deficient only for CA VB."

3. GWAS: have any phenotypic traits been linked to SNPs in the region of the CA VA or CA VB genes?

Our response: We have added the following paragraph to the end of "Physiological functions of CA VA and CA VB" section (page 12): "GWAS Central database (<https://www.gwascentral.org/index>) can be searched to find genome-wide associations between single nucleotide polymorphisms (SNPs) and phenotypes. Using p-value threshold - $\log p \geq 2$, there are two phenotypes significantly associated with markers in gene or region of the CA5A gene. The phenotype ontology annotation indicates a linkage to Alzheimer's disease and breast cancer. At this p-value threshold, the CA5B gene shows no phenotypic associations."

Referee #2:

The paper by Aspatwar et al. entitled "Mitochondrial carbonic anhydrase VA and VB: Properties and roles in health and disease" is an interesting review regarding these two mentioned mitochondrial isoforms. The review is very well written and dissects all the aspects of these two isoforms, from the biochemical to the functional features and the inhibition studies carried out on the two enzymes.

The presented review gives complete information on recently progress made in the field.

Only some minor revisions are necessary:

- 1) Abstract figure: Rephrase last two lines regarding presence of CA VA and B, specify which are the organs where CA VB is present.

Our response: We thank the reviewer for this comment. We have modified the text for Abstract figure as follows: " The liver hepatocytes are the main source of carbonic anhydrase VA with weaker signals in brain, testis, and muscle. The VB isozyme is more widely spread in several organs, such as brain, heart, liver, lung, kidney, spleen, intestine, testis, muscle, and pancreas."

We also defined the corresponding section more precisely in Introduction: "CA VA is mainly expressed in the liver, with some expression also observed by Western blot in the brain, testis, and skeletal muscle (Shah et al., 2000). The CA VB isozyme has a much wider tissue distribution and is expressed in the heart, liver, lung, spleen, intestine, pancreas, testis, skeletal muscle, kidney, salivary gland, brain, and spinal cord, suggesting different physiological roles for these two mitochondrial isozymes (Fujikawa-Adachi et al., 1999; Ghandour et al., 2000; Shah, et al., 2000)."

- 2) Pag 4, lane 14, an "and" is missing between VA VB.

Our response: Instead of adding "and" we replaced VA and VB with "CA activity" which is a more accurate term in this context (page 4).

- 3) Pag 5, Biochemistry CA VA and CA VB. It would be clearer to the reader if authors would insert a table regarding the catalytic efficiency of these two isoforms compared to the other isoforms, rather than listing them in the text.

Our response: We have added Table 1 according to the reviewer's suggestion.

- 4) Pag 6 and 7, Authors repeat twice the information regarding the amino acid similarity of CA VB to CA VA (66%). Please edit.

Our response: We thank the reviewer for pointing out this repeated sentence. The first repeated sentence has been removed.

- 5) Pag 7, Authors should include in the text the information that when protein numbering is not inclusive of the signal peptide, residue position is annotated with a + before the residue number. This can help the reader and avoid confusion when numbered residues are cited within the manuscript.

Our response: The information requested by the reviewer has been added to the text in the "Molecular biology of CA VA and CA VB" section (page 8).

- 6) Pag 7, CA VB contains 6 cysteines and not 5 as mentioned in the text. Please correct the sentences in the text.

Our response: We thank the reviewer for pointing out this error. The sentence about these cysteines has been corrected (Figure 3 legend).

- 7) Pag 8, Fig 3, please box the cysteine in position 92 of CA VA and VB, and edit Figure legend.

Our response: Referring to the previous comment, the sentence and figure has been corrected accordingly.

- 8) Pag 7 Authors state that CA VB may contain four Cys residues possibly involved in intradisulfide bond formation referring to reference 39 which is a PNAS manuscript on CA IV. Please note that in CA IV the cysteines involved into disulfide bonds are not conserved in CA VB. Please check the reference. Moreover, the crystallographic structure of murine CA V was solved showing that all the cysteine residues are reduced. Why Authors do not use this structure to predict the CA VB redox state of cysteines? Please check

Our response: We thank you the reviewer for pointing out the incorrect information about the cysteine residues of CA VB. We agree that the correlation to CA IV is not justified.

Therefore, we deleted the incorrect statement and added the following (on page 8):

"Although human CA VA and VB contain four and six cysteine residues, respectively, no disulfide bridges as post-translational modifications have been reported in these enzymes (Di Fiore et al. 2020)."

Dear Professor Parkkila,

Re: JP-TR-2022-283579R1 "Mitochondrial carbonic anhydrase VA and VB: Properties and roles in health and disease" by Ashok Aspatwar, Claudiu T. Supuran, Abdul Waheed, William S Sly, and Seppo Parkkila

We are pleased to tell you that your paper has been accepted for publication in The Journal of Physiology.

Authors should note that it is too late at this point to offer corrections prior to proofing. The accepted version will be published online, ahead of the copy edited and typeset version being made available. Major corrections at proof stage, such as changes to figures, will be referred to the Editors for approval before they can be incorporated. Only minor changes, such as to style and consistency, should be made at proof stage. Changes that need to be made after proof stage will usually require a formal correction notice.

All queries at proof stage should be sent to: TJP@wiley.com

Yours sincerely,

Professor Laura Bennet
Senior Editor
The Journal of Physiology
<https://jp.msubmit.net>
<http://jp.physoc.org>
The Physiological Society
Hodgkin Huxley House
30 Farringdon Lane
London, EC1R 3AW
UK
<http://www.physoc.org>
<http://journals.physoc.org>

P.S. - You can help your research get the attention it deserves! Check out Wiley's free Promotion Guide for best-practice recommendations for promoting your work at www.wileyauthors.com/eoo/guide. You can learn more about Wiley Editing Services which offers professional video, design, and writing services to create shareable video abstracts, infographics, conference posters, lay summaries, and research news stories for your research at www.wileyauthors.com/eoo/promotion.

IMPORTANT NOTICE ABOUT OPEN ACCESS: To assist authors whose funding agencies mandate public access to published research findings sooner than 12 months after publication The Journal of Physiology allows authors to pay an Open Access (OA) fee to have their papers made freely available immediately on publication.

You can check if your funder or institution has a Wiley Open Access Account here: <https://authorservices.wiley.com/author-resources/Journal-Authors/licensing-and-open-access/open-access/author-compliance-tool.html>

EDITOR COMMENTS

Reviewing Editor:

Thank you for submitting your revised commissioned review article titled "Mitochondrial carbonic anhydrase VA and VB: Properties and roles in health and disease" to The Journal of Physiology. The two external peer reviewers and the academic

reviewing editor believe the changes made in response to reviewers' critiques have improved the manuscript greatly. Thank you.

REFEREE COMMENTS

Referee #1:

The authors addressed all my concerns adequately.

Referee #2:

The paper " Mitochondrial carbonic anhydrase VA and VB: Properties and roles in health and disease" can now be accepted for publication.

1st Confidential Review

20-Nov-2022